# Bearing fault diagnosis method based on enhanced VMD and adaptive-optimized SDAE

**Xianlin Ren[1,2]⊛, Haowen Li[1]⊛\*, Laixian Chen[1]⊛, Siyao Xiong[1]⊛, Zhengwen Li[1]⊛**

**1** School of Mechanical and Electrical Engineering, University of Electronic Science and Technology of China, Chengdu, Sichuan, China, **2** Institute of Electronic and Information Engineering of UESTC in Guangdong, Dongguan, Guangdong, China

⊛ These authors contributed equally to this work.
\* 202322040545@std.uestc.edu.cn

## Abstract

Motor rolling bearing is a fundamental component of industrial production, and its vibration signal extraction and fault diagnosis are challenging because of the effect of operating characteristics and external noise. This research initially proposes an adaptive variational mode decomposition approach based on dung beetle optimization algorithm to decompose and extract signals. At the same time, a composite optimization indicator function based on Tanimoto coefficient, permutation entropy and kurtosis are presented as the fitness function of decomposition to increase the flexibility and robustness of the technique. Next it combines with composite multiscale permutation entropy to finish feature extraction and create feature vectors. Finally, an enhanced inertia weights and Cauchy chaotic mutation-Sine Cosine Algorithm is utilized to optimize the hyperparameters of the stacked denoising auto-encoders network and construct a fault diagnosis model. The CWRU open bearing dataset is used to comprehensively evaluate the performance of the method, and the experimental results will be compared to show that the method proposed in this paper can effectively extract signal features in the situation of strong noise, while ensuring a high prediction accuracy, and has stronger adaptability and noise resistance compared with other methods.

## 1. Introduction

Rolling bearings are the key parts of most mechanical transmission devices, and due to the intensity of their work they are more prone to failures. The damage of the bearing can cause economic and social losses, or even threaten the safety of the personnel [1–3]. Therefore, bearing fault diagnosis is a vital aspect of system design and maintenance since it helps to increase production efficiency resulting in lower costs and accidents [4].

**Data availability statement:** All bearing signal files are available from the Case Western Reserve University Bearing Data Center. The relevant information and details can be found in the official website: https://engineering.case.edu/bearingdatacenter/welcome The download URL of the dataset and the description of it is followed: https://engineering.case.edu/bearing-datacenter/download-data-file we have updated the research data in the public repository Kaggle which is in the list of recommended repositories and additional information on PLOS standards for data deposition., the relevant URL is as follows: https://kaggle.com/datasets/fa1491856b927e78244555d7456b73e-6455cee523f3154dbdf02c3d91d06ed0e.

**Funding:** The research is supported by the National Natural Science Foundation of China(51875090. The funders played roles in decision to publish and preparation of the manuscript.

**Competing interests:** The authors have declared that no competing interests exist.

Due to the complex and changing working circumstances in the real operating environment, the rolling bearing vibration signal shows the characteristics of non-linearity, non-stationarity, weak periodicity and poor signal-to-noise ratio. When rolling bearing failure occurs, its vibration signal comprises a significant number of motion state signals, which are non-smooth and multi-component modulated signals [5,6]. Therefore, when a rolling bearing fails, feature extraction of the effective information of its vibration signal is a vital step in recognizing and assessing the bearing failure [7,8]. However, in the collected vibration signals, the fault features are not evident concealed in a vast quantity of data, and these signals are frequently inevitably mixed with a considerable amount of noise, which interferes with the results of feature extraction. Therefore, how to extract features with useful information from vibration signals affected by noise and develop feature vectors with appropriate noise immunity and robustness is the primary challenge in the feature extraction stage of bearing a fault diagnosis. Various kinds of modal decomposition algorithms are commonly employed for signal feature extraction [9–12]. Among them, DRAGOMIRETSKIY [13] proposed variational mode decomposition (VMD) in 2014, which decomposes the signal into several sub-signals with specific frequencies and limited bandwidth, and effectively solves the end-point effect and mode stacking problems that are prone to occur in modal decomposition. However, the decomposition and extraction impact of VMD is influenced by the preset decomposition parameters, and if the selection is based on expert experience or a priori knowledge, it will result in a restricted application of VMD and lack of appropriate adaptivity. Meng et al. optimized VMD method by an adaptive spectrum segmentation and validated in bearing fault diagnosis [14], Huang et al. optimized the parameters of VMD in terms of signal energy loss [15], Lin et al. [16] conducted gear fault diagnosis by using cuckoo search (CS) to optimize VMD and probabilistic neural network. According to the current commonly used methods for VMD signal decomposition, only the theoretical features obtained from signal decomposition are obtained, and there is a lack of evaluation indicator to judge whether there is any omission or duplication in feature extraction. The effective information in the decomposed signal characteristics is restricted. Therefore, in order to further increase the usefulness of the feature information of the decomposed signal, a new function *L* is presented in this study. This function *L* which incorporates permutation entropy value, average kurtosis and Tanimoto coefficient etc., evaluates the decomposed signal through the fitness function composed of multiple feature indicators to reduce the omission of effective feature information, which makes the signal decomposition process have higher extraction efficiency, accuracy as well as stronger robustness and noise immunity. It eventually improves the preset decomposition parameters in the VMD approach by employing the dung beetle optimizer (DBO). Compared with the traditional optimization approach, the parameter optimization of VMD decomposition applying DBO may greatly boost its operating efficiency and convergence accuracy.

Bearing working environment is complex, the collected signals are also mixed with a large number of environmental noises, in order to further improve the feature extraction of vibration signals, to adopt entropy algorithm is the current research

hotspot. Permutation entropy (PE) [17] is a technique for identifying randomness and complexity in one-dimensional time data. Through coarse-grained processing, the original time series can be divided into several short time series. The coarse-grained time series can represent the dynamic distribution characteristics of the original signal at a certain scale [18]. So composite multi-scale permutation entropy (CMPE) is proposed by placing multiple average filters on the time series to complete the multi-scaling, and joining the time-shift operation on its basis, which greatly improves the coarse-grained degree of PE, and better ensures the extraction of signal features. It may substantially alleviate the issue of inadequate coarse granularity of permutation entropy, and better assure the reducibility and richness of the retrieved signal characteristics. In this research, we will utilize the VMD approach in conjunction with CMPE to extract signal features and create the input feature matrix to optimize the adequacy, reducibility and accuracy of feature extraction. The reconstructed feature vector using CMPE can better reflect the feature information and ensure the consistency between the features and the original signal, especially CMPE has a higher sensitivity to the abnormal changes in the vibration signal, which is conducive to the feature extraction of the fault signal.

The quantity of vibration signal data is vast, the useful information is difficult to be recovered, and the models constructed based on manual experience typically have the issue of inadequate generalization. For the still vast quantity of feature information, how to develop efficient and reliable prediction models is one of the focus challenges. In recent years, deep learning and machine learning have arisen and demonstrated considerable promise in the field of bearing fault diagnosis. With the capacity of successfully extracting features from signals while not needing expert knowledge, deep learning methods may overcome the disadvantages of the conventional approach in signal-based fault diagnosis [19–21]. Such as convolutional neural networks (CNN) [22], recurrent neural networks (RNN) [23], deep belief networks (DBN) [24], etc. Among them, auto-encoders (AE) [25] is one of these unsupervised deep learning networks. And stacked denoising auto-encoders (SDAE) is proposed by stacking several AE networks and introducing noise. For SDAE network, finding the optimal network hyperparameters is a key step to optimize the performance of the model. Duan [26] et al. utilized bat algorithm (BA) to optimize the network parameters of SDAE, which was used to anticipate the wind power and avoid failures; Wu [27] et al. used particle swarm optimization (PSO) with adaptive weights improve the structure of SDAE and to categorize gearbox defects. Xu et al. proposed a technique of fault extraction for motor of electric vehicle by integrating SDAE with support vector machine (SVM) classifier [28]. Although the above methods are optimized for the selection of hyperparameters of SDAE network, they are easy to cause local optimum of parameters, and there is still room for improvement for the global space search. Therefore, this paper proposed to utilize the improved inertia weights and Cauchy chaotic mutation-Sine Cosine Algorithm (IWCCSCA) [29] to optimize the structure and parameters of SDAE which will address the issue of local optimal and global optimization efficiency in the standard SCA parameter optimization method, and increase the generalization and prediction ability of SDAE networks.

## 2. Materials and methods

### 2.1 Principle of VMD

VMD is an excellent adaptive time-frequency signal decomposition method, which estimates the components of each signal by solving a frequency-domain variational optimization problem. VMD assumes that all the components are narrow-band signals concentrated around their respective center frequencies, so VMD builds up a constrained optimization problem based on the conditions that the components are narrow-banded, so as to estimate the center frequencies of the signal components and reconstruct the corresponding components. VMD is underpinned by a sound mathematical theory and is essentially an adaptive optimum Wiener filter [30].

The feature decomposition of the signal is accomplished by constructing the following variational problem. And the constraint is that the sum of all modals is equal to the original signal $f(t)$.

$$\min_{\{u_k\}\{\omega_k\}} \left\{ \sum_k \left\| \partial_t [(\delta(t) + \frac{j}{\pi t}) * u_k(t)] \cdot e^{-j\omega_k t} \right\|_2^2 \right\}$$
$$s.t. \quad \sum_k u_k(t) = f(t)$$

$$(1)$$

where $u_k(t)$ is the intrinsic mode function (IMF), which is essentially an FM signal, and $\omega_k(t)$ is the center frequency of $u_k(t)$.

In order to solve the above variational problem, a quadratic penalty factor α and Lagrange operator λ(t) are introduced into the Lagrange multiplier method, which transforms the constrained variational problem into an unconstrained variational problem, and an augmented Lagrange expression is obtained.

$$L(\{u_k\}, \{\omega_k\}, \lambda(t)) =$$
$$\alpha \sum_k \left\| \partial_t [(\delta(t) + \frac{j}{\pi t}) * u_k(t)] \cdot e^{-j\omega_k t} \right\|_2^2 + \left\| f(t) - \sum_k u_k(t) \right\|_2^2 + \left\langle \lambda(t), f(t) - \sum_k u_k(t) \right\rangle$$

$$(2)$$

Then the cross-multiplier method is utilized to constantly update the iterations $u_k(t)$, $\omega_k(t)$, λ(t), until convergence to the requisite precision, finishing the signal decomposition and feature extraction.

## 2.2 Method of optimization

In the signal decomposition of the above VMD methods, the number of channels of the IMFs $K$ and the quadratic penalty factor $\alpha$ are the key elements of the decomposition objectives and effects. Usually, they are set based on a priori knowledge or expert experience, but such methods lack adaptivity and are susceptible to human subjective factors, so it is proposed to use the emerging intelligent optimization algorithm, dung beetle optimization (DBO) [31], to optimize the VMD parameters. In the process of parameter optimization, an appropriate fitness function should be chosen as the optimization aim of the process to assure the correctness and efficacy of the signal decomposition findings. The general single-indicator fitness function has a certain applicability, but the optimization results are only for a single type of feature indicator, and the results have not been mathematically verified in multiple indicator dimensions, which lacks sufficient adaptivity and robustness in the face of a variety of feature types of signals. In order to expand the applicability range of the fitness function, strengthen the application breadth of this optimization process and the information entropy value of the decomposition results, a composite optimization indicator function $L$ is proposed to replace the single fitness function based on the perspective of multi-dimensional data fusion, and the DBO-VMD method is further optimized and validated. In the parameter optimization process of DBO-VMD, a specific fitness function $L$ is constructed as an evaluation criterion, in which Tanimoto coefficient ensures signal consistency before and after the decomposition of vibration signals, permutation entropy extracts unusual changes in vibration signals caused by faults, and kurtosis detects changes in the intensity of the impact of the signals, which is one of the characteristics of fault signals. Therefore, the fitness function may successfully extract the features, such that the extracted features have more and richer information, and minimize the absence of information linked to fault characteristics.

Tanimoto coefficient is utilized to indicate the similarity of the data before and after the decomposition, if it is the closer to 1, it indicates that the greater the degree of similarity. Here it is introduced into the function can characterize the degree of similarity before and after the decomposition of vibration signals, to ensure that the retention of the key information, the formula is shown in Equation (3).

$$T_c = \left| \frac{\sum_{i=1}^N x_i(t)y_i(t)}{\sum_{i=1}^N x_i^2(t) + \sum_{i=1}^N y_i^2(t) - \sum_{i=1}^N x_i(t)y_i(t)} \right|$$

$$(3)$$

where $T_c$ is the Tanimoto coefficient, $x_i(t)$, $y_i(t)$ are the time series data before and after the input and output respectively.

The value of permutation entropy of each component can effectively map the degree of change and complexity of the time series, and has a strong sense of the minor changes in the mechanical system response signal [32]. At the same time, the bigger the value of PE, the more chaotic the time series data distribution is. So it is one of the main factors for fault diagnosis.

$$H_{PE} = \frac{\left(-\sum_{i=1}^{k} P_i \cdot \lg(P_i)\right)}{\lg(m!)} \qquad (k \leq m!)$$

(4)

where $H_{PE}$ is the value of PE, $m$ denotes the number of embedding dimensions used to reconstruct the time series, and $P_i$ denotes the probability of occurrence of each component sequence alignment in the reconstructed vector. Thus from the preceding description we may deduce that the bigger the value of PE, the more prominent and chaotic the raw vibration signal fluctuation is, which is most likely owing to the fault.

kurtosis value is a measure of the peak state of the distribution of random variables dimensionless parameter, which may present the concave and flat degree of the top of sample function, therefore it can be used to determine that the shock signal strength. The bigger the value of kurtosis, the more irregular and powerful the bearing vibration shock is, which suggests that the bearing may be wrecked.

$$K_u = \frac{1}{N}\sum_{i=1}^{N}\left[\frac{x(t)-\bar{x}}{\sigma}\right]^4$$

(5)

where x(t) is the IMF signal, $\bar{x}$ is the mean value of the IMF signal, and σ is the standard deviation of the IMF signal.

Based on the parameters previously discussed, the $L$ function can be created:

$$L = T_c\left(a \cdot H_{PE} + b \cdot K_u\right)$$

(6)

where $a$ and $b$ are used as the weighting factors of the permutation entropy and the kurtosis, and $a + b = 1$. Based on the characteristics of kurtosis and permutation entropy in the fault vibration signal, for the faulty bearing vibration signal, the permutation entropy is more sensitive to the degree of integrity of the overall signal, and the kurtosis is for the characterization of faulty impact signals, so this method is to take $a = 0.6$, $b = 0.4$ as the final indexes.

From the above description we can learn that the higher value of $L$ represents the more valid information it contains and the closer it is to the initial signal, which means the bearing which is the source of the time series is more likely to be wrecked. So we can derive the fitness function for use in the optimization of $K$ and α in VMD:

$$Fitness\{K, \alpha\} = \min\left(-\frac{\sum_{i=1}^{K} L}{K}\right)$$

(7)

According to Equation (7) we know that for the fitness function of VMD parameter optimization, it is a minimization problem. Therefore the minimization of the fitness function will be employed as the optimization target of the DBO.

Combined with the above composite indicator fitness function method, the specific steps of applying DBO to the adaptive parameter optimization of VMD method are shown below:

(1) Presetting the starting parameters of the DBO algorithm. The overall number of dung beetle populations is set to be 30, and the numbers of each behavioral population are 6 for rolling, 6 for breeding, 7 for foraging, and 11 for stealing, respectively.

(2) The VMD parameters that need to be optimized in this research are two parameters: the number of IMF channels $K$, and the quadratic penalty factor $\alpha$, therefore the two-dimensional vector $X = [X_1, X_2]$ is set as the population unit, and $X_1$, $X_2$ stand for $K$, $\alpha$ accordingly. Based on the a priori knowledge and experience, the value range of $X_1$ is [3,8], while the value range of $X_2$ is [300,5000].

(3) $H_{PE}$ and $K_u$ in Equation (7) are normalized and the $X$ that minimizes the fitness function is computed as the starting population location.

(4) Through the four behavioral populations of diverse ways for repeated iterations to update the population position, and pick reducing *Fitness* as the ideal selection criterion for the population.

(5) If the current number of iterations $t$ is less than the predefined maximum number of iterations $T$, return to (4) to continue the execution, and vice versa, output $X$ as $K$, $\alpha$.

## 2.3 Construction of feature vector

The CMPE algorithm improves the MPE algorithm's insufficient degree of coarse-graining through the composite coarse-graining method, and obtains multiple aligned entropy values for different time series, and uses the average value as the entropy value of the time series, so that the effectiveness and robustness of the feature extraction can still be maintained under the large-scale factor, and the feature vectors used for inputting into the deep learning network are constructed by combining this method and the above mentioned DBO-VMD method. The particular stages of the procedure are mentioned below:

(1). For the original data sequence $X = \{x(i), i = 1, 2, 3, \ldots, N\}$, let it through the process of coarse-graining, the scale factor is preset to be $\tau$, and the process is shown in Equation (8).

$$y_{k,j}^{(\tau)} = \frac{1}{\tau} \sum_{i=(j-1)\tau+k}^{j\tau+k-1} x(i) \quad 1 \leq j \leq \left[\frac{N}{\tau}\right], \quad 1 \leq k \leq \tau$$

(8)

where $y_{k,j}^{(\tau)}$ denotes the $j$th sampling point of the $k$th coarse-grained sequence under the scale factor $\tau$, and [] denotes a rounding operation that ensures that the sampling point is integer.

(2). For each coarse-grained sequence $y_{k,j}^{(\tau)}$, compute the alignment entropy value $H_{PE}$ within the sequence, and then average the alignment entropy value under the scale factor $\tau$ as the output entropy value, as shown in Equation (9).

$$H_{CMPE}(X, \tau, m, \lambda) = \frac{1}{\tau} \sum_{k=1}^{\tau} H_{PE}\left(y_k^{(\tau)}, m, \lambda\right)$$

(9)

where $m$ is the embedding dimension used to rebuild the time series and $\lambda$ denotes the time lengthening in the reconstruction procedure. $H_{PE}$ is the value of PE, which has been defined in Equation (4)

(3) Use the $K$ and $\alpha$ acquired from the optimization approach as the preset parameters of the VMD method to decompose the original signal, and finish the first feature extraction, then we may get many IMFs having independent features whose total number is $K$. Next use the optimization indicator function to screen and extract the decomposed IMFs. Calculate the results of fitness function $L_i$ of each IMF and the mean value $\bar{L}$ of all, and the filtering rule is $L_i \geq \theta_L$, then the IMF that is considered as the $i$th component which is legitimate, and vice versa is invalid. For the threshold coefficient, $\theta_L$ is originally selected as 0.75, which may assure the efficacy of feature extraction.

(4) For different samples, the number of valid IMFs $p_i$ varies, and in order to keep the input dimensions consistent, a "leveling" strategy is adopted. Assuming that the number of preset component IMFs is $p$, if $p_i < p$, then the value of $\theta$ will be progressively lowered, and vice versa, the value of $\theta$ will be gradually raised until the number of output IMFs equals $p$.

(5) The original signal of a single sample is set to extract $q$ sequences for feature extraction, thus the dimension of the feature vector of a single sample is $p \cdot q$, and this feature vector is utilized as the input vector for future input to the SDAE network for training.

Table 1 shows the pseudocode of the method.

**Table 1. Feature vector extraction method based on CMPE and optimized VMD.**

| |
|---|
| **Algorithm 1** Feature vector Extraction method based on CMPE and optimized VMD |
| **Input**: Raw signal data sequence $X = \{x(i), i = 1, 2, 3, \ldots, N\}$. |
| **Output**: Feature vector $\mathbf{V} \in R^{p*q}$. |
| 1:  $X = \{x(i), i = 1, 2, 3, \ldots, N\}$//Input raw signal data; |
| // Step 1. Optimized VMD decomposition based on DBO |
| 2:  $Para = [K, \alpha] = DBO [min (Fitness)]$; |
| 3:  $[IMF_1, IMF_2, IMF_3, \ldots] = VMD (X, K, \alpha)$; |
| // Step 2. Calculation of CMPE |
| 4:  Coarse-Graining of time sequence according to Eqs. (8); |
| 5:  Compute the value $H$ of CMPE according to Eqs. (9); |
| // Step 3. Feature extraction based on IMF and function $L$ |
| 6:  **for** ($i = 1$ to $K$) |
| 7:  $L_i = L (IMF_i)$// according to Eqs. (6) |
| 8:  **end for** |
| 9:  $L_0 = mean [sum (L_i)]$; |
| 10:  **for** ($i = 1$ to $K$) **do** |
| 11:  **if** ($L > \theta \cdot L_0$) **then** $IMF_{valid} \leftarrow IMF_i$; end if |
| 12: **end for** |
| 13: $p_0 = p$, $q_0 = q$, $p_1 = \| IMF_{valid} \|$;// p: the expected number of IMFs, q: the number of extracted sequences in every IMF, $p_1$: the actual number of IMFs |
| 14: **while** $p_1 \neq p_0$ **do** |
| 15:  **if** ($p_1 > p_0$) **then** ($\theta = \theta + \Delta_0$);// $\Delta_0$: Step size |
| 16:  **else** ($\theta = \theta - \Delta_0$) |
| 17:  **end if** |
| 18: **end while** |
| 19: **for** ($j = 1$ to $q_0$) |
| 20:  $H_j = CMPE(Y = \{y(i)\})$// $Y = \{y(i)\} \in IMF_n \in IMF_{valid}$ |
| 21: **end for** |
| // Step 4. Feature vector construction |
| 22: $V = zeros (p_0, q_0)$ |
| 23: **for** (i = 1 to $p_0$) **do** |
| 24:  **for** (j = 1 to $q_0$) **do** |
| 25:  $\mathbf{V} [i, j] = H_j$ |
| 26:  **end for** |
| 27: **end for** |
| 28: return $\mathbf{V} \in R^{p*q}$ |

## 2.4 IWCCSCA

The SCA generates the initial population position by means of a randomized scale division over the range of intervals, and iteratively updates the position by Equation (10):

$$x_{i,j}^{(t+1)} = \begin{cases} x_{i,j}^{(t)} + \sin r_2 \cdot r_1 \cdot \left| r_3 \cdot x_{g,j} - x_{i,j}^{(t)} \right| & r_4 < 0.5 \\ x_{i,j}^{(t)} + \cos r_2 \cdot r_1 \cdot \left| r_3 \cdot x_{g,j} - x_{i,j}^{(t)} \right| & r_4 \geq 0.5 \end{cases}$$
$$r_2 \in [0, 2\pi], \quad r_3 \in [0, 2], \quad r_4 \in [0, 1],$$

(10)

Where $x_{i,j}^{(t)}$, $x_{i,j}^{(t+1)}$ are the position of individual of the $j$th dimension before and after the iterative update, respectively, and $x_{g,j}$ is the component of the current global optimal solution in the $j$th dimension. $r_2$, $r_3$, and $r_4$ are all uniformly distributed random variables. and $r_1$ is the amplitude conversion factor as shown in Equation (11) shown. The principle of SCA is briefly shown in Fig 1.

$$r_1 = a - a \cdot \frac{t}{T_{max}}$$

(11)

where $T_{max}$ is the maximum number of iterations, $t$ is the current iteration number, and $a$ is a constant, which is selected as $a = 2$ here.

Although the above classic SCA approach has some optimization ability, it has poor global search ability and sluggish convergence time when addressing difficult multi-dimensional optimization issues. Therefore, it can be improved according to the amplitude adjustment factor update, inertia weight update, and Cauchy chaotic variability principle, which significantly improves the performance of the original SCA algorithm by adapting the optimization adjustment of multiple parameters in the iterative updating formula of the SCA method. The performance of the original SCA algorithm is significantly improved in terms of global optimization accuracy and convergence speed.

First, $r_1$ in the original technique is utilized to balance the weight between global search and local optimization. However, the initial $r_1$ value is updated depending on the number of iterations without considering the current population

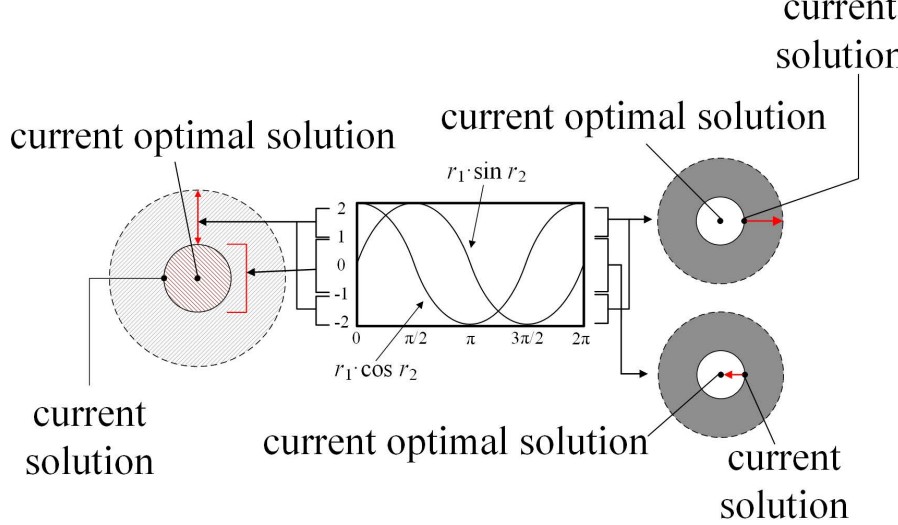

**Fig 1. Principal of SCA.**

location and the overall optimization search method. If $r_1$ is too big, the global search ability is powerful, but it will result in inadequate search, which is easy to cause the local optimization; If $r_1$ is too small, the iterative step size will become smaller appropriately, which leads to the inability to converge rapidly. Therefore, an adaptive updating approach of amplitude adjustment factor based on exponential function will be employed, as illustrated in Equation (12):

$$r_1^{(t)} = a - a \cdot [(e^{t/T_{max}} - 1)/(e - 1)]^k \tag{12}$$

Where $e$ is a natural constant and $k$ is a conditioning factor. In Equation (12), compared to the original approach, this $r_1$'s value declines slowly at the beginning, increasing the number of searches and improving the searching ability, also it speeds the reduction at a later stage, making the algorithm converge rapidly.

Secondly, it is the position update based on inertia weights. Referring to the update method of particle swarm optimization algorithm, the inertia weights are increased in the early stage to strengthen the global search ability, and the proportion of inertia weights is reduced in the later stage to improve the local optimization ability and accelerate the convergence. Therefore, the better inertia weight updating technique is indicated in Equation (13):

$$w^{(t)} = w_{min} + (w_{max} - w_{min}) \exp\left[-\frac{1}{2} \times \left(\frac{t}{T_{max}}\right)^2\right] \tag{13}$$

where $w^{(t)}$ is the current update inertia weight; $w_{max}$ is the initial inertia weight, i.e., the maximum value of inertia weight; and $w_{min}$ is the ending inertia weight, i.e., the minimum value of inertia weight. After introducing the inertia weight $w^{(t)}$ variable, the position update method is Equation (14):

$$x_{i,j}^{(t+1)} = \begin{cases} w^{(t)} x_{i,j}^{(t)} + \sin r_2 \cdot \bar{x} & r_4 < 0.5 \\ w^{(t)} x_{i,j}^{(t)} + \cos r_2 \cdot \bar{x} & r_4 \geq 0.5 \end{cases} \tag{14}$$

Finally, it is to add mutation perturbation to avoid the population from sliding into local optimization. This technique employs a hybrid Cauchy chaotic variation method to disturb the optimum solution $x_g$ by adding Cauchy variation individuals to the ideal solution $x_g$ and combining it with the usage of chaotic systems to boost the population diversity and broaden the search space.

Then the Cauchy operator is added into the optimal solution to bring it out of local optimality, as illustrated in Equation (15):

$$x_{new} = x_g + Cauchy \cdot x_g \tag{15}$$

where $x_g$ is the current optimal solution of the original method, which can also be considered as an elite individual, Cauchy is the Cauchy operator that conforms to the Cauchy distribution, and $x_{new}$ is the new elite individual after the resultant Cauchy variation is updated.

The chaotic variational system utilizes Logistic chaotic values, which are defined as illustrated in Equation (16):

$$\varphi^{(t+1)} = c \cdot \varphi^{(t)} \cdot \left(1 - \varphi^{(t)}\right) \tag{16}$$

where $c$ is the chaos coefficient, generally selected as $c = 4$.
To recap, the individual Cauchy chaotic variational perturbation is defined as follows.

$$x_{new} = \begin{cases} x_g + Cauchy \cdot x_g & r_5 > 0.5 \\ x_{min} + \varphi \cdot (x_{max} - x_{min}) & r_5 \leq 0.5 \end{cases} \tag{17}$$

where $r_5$ is the Cauchy chaos weight coefficient, which is used to select the mutation method and takes the value of [0,1] as a random number.

## 2.5 Model of diagnosis

Stacked denoising auto-encoder is based on the creation of neural network Auto-Encoder (AE), and based on the AE network, Vincent et al. suggested [33] Denoising Auto-Encoder (DAE), which replicates the condition of the data being interfered with by noise by adopting a random zero-setting of the input data using the approach of Drop-out-like method. and stack many DAEs then we may acquire the SDAE with better generalizability and robustness.

In SDAE network, the principle of sparse coding is commonly utilized to establish a sparse parameter ρ so that the average activation value of the hidden layer node j is near to ρ, which fulfills the objective of capturing the critical information of the input while deleting part of the redundant information. Based on this notion, the loss function of this network model is:

$$J(W, b) = \frac{1}{N} \sum_{i=1}^{N} \left( \frac{1}{2} \left\| x^{(i)} - y^{(i)} \right\|^2 \right) + \frac{\lambda}{2} \sum_{l=1}^{L} \sum_{m=1}^{M} \sum_{n=1}^{N} \left( W_{mn}^{(l)} \right)^2 + \beta \sum_{j=1}^{S} KL \left( \rho | \rho_j \right)$$

(18)

where $x^{(i)}$, $y^{(i)}$ are the input and the output of the network, respectively. The first term denotes the reconstruction loss from input data to output data; and the second term is the regularization loss, which is used to prevent the network model from overfitting. and *KL* scattering serves as a penalty term to constrain the activation values around ρ. β denotes the sparse factor weight, in this paper, we take β = 3, and the specific calculation formula is shown below:

$$KL \left( \rho | \rho_j \right) = \rho \ln \frac{\rho}{\rho_j} + (1 - \rho) \ln \frac{1 - \rho}{1 - \rho_j}$$

(19)

In SDAE networks, the design and values of plural hyperparameters directly affect the accuracy, noise immunity, generalizability and other properties of the final model. Currently, most of the SDAE hyperparameters are determined by empirical enumeration of multiple combinations to obtain a better set of hyperparameters, which need to be specially tested and selected for specific fault diagnosis problems. For fault classification tasks in diverse domains, the suggested hyperparameters show limited generalization performance [34]. In order to boost the generalization capacity of the network and strengthen the resilience of the model in the face of complex disturbances, a set of network hyper-parameters for SDAE are determined via adaptive optimization using the IWCCSCA approach. By tuning the hyperparameters of the SDAE network with IWCCSCA, the model's generalization ability may be strengthened, while simultaneously obtaining higher resilience and interference resistance.

For deep learning network models, the more hidden layers, the greater accuracy of its output, however, but, when the number of hidden layers exceeds 4, the model generalization capacity will exhibit a considerable decline [35]. Therefore, the number of hidden layers of the SDAE network in this paper is chosen to be 3. Therefore, this paper needs to optimize 7 network hyperparameters of SDAE: 3 hidden layer nodes numbers, 3 hidden layer sparsity coefficients, and 1 input data zero-ratio. Therefore the stages for developing the SDAE network model optimized based on the IWCCSCA approach are described below:

(1) Setting up the target population, from the above elaboration we know that the network has seven hyperparameters to be optimized, so assume that each single population is a seven-dimensional vector $Xi = [x_{i1}, x_{i2}, \ldots, x_{i7}]$, where $x_{i1}$, $x_{i2}$, $x_{i3}$ are the number of nodes in the three hidden layers, $x_{i4}$, $x_{i5}$, $x_{i6}$ are the sparsity coefficients corresponding to the three hidden layers, and $x_{i7}$ is the proportion of the zeroes in the input data.

(2) Set the population size $N = 20$, the number of iterations is selected as $T_{max} = 100$, select a suitable target classification error rate $R_{error}$, initialize the population according to the range of values of each corresponding component of the population, and calculate the corresponding $R_{error}$ for each individual.

(3) Updating of stock holdings according to the method in section 2.4.

(4) Calculate the updated population individual corresponding to $R_{error}$, and locate the population individual corresponding to the least $R_{error}$.

(5) Check whether the iteration reaches the convergence condition, if the minimum value of $R_{error}$ is less than the set threshold or the number of iterations $t$ is equal to the set maximum number of iterations $T_{max}$, then it is considered that the optimization objective is reached, and exit the loop, and vice versa, repeat the step (3).

(6) The population individuals corresponding to the smallest $R_{error}$ achieved in step (5) are utilized as the optimization results of the SDAE hyperparameters, and the results are fed to the network hyperparameters to complete the network model.

The following Fig 2 depicts the model.

## 2.6 Introduction of data

The faulty bearing vibration signal dataset utilized for the experiment is a publicly available dataset from Case Western Reserve University (CWRU). It is an open data set, which is a benchwork dataset and frequently used in fault diagnosis and it can be acquired via the website: https://csegroups.case.edu/bearingdatacenter. The bearing type of the test rig is deep groove ball bearing, and the motor bearing model is SKF-6205, which is processed by EDM technology to mimic the failures of four different sizes of 0.007", 0.014", 0.021", and 0.028", respectively. The sampling frequency of the experimental data was classified into 12kHz and 48kHz, and the data categories include Normal, Inner Race fault, Rolling ball fault, and Outer Race fault. The final data collection collected the acceleration time series data of the bearings of this experimental platform under load conditions of 0 hp, 1 hp, 2 hp, and 3 hp.

Based on the real experimental demands, a total of 10 rolling bearing data types with load circumstance of 0 hp will be chosen, and their comprehensive information is provided in Table 2.

For the selected 10 groups of bearing vibration signals, the signal waveforms corresponding to different fault types are significantly different, and the features are complex, so the extraction of features is especially critical to complete the differentiation and diagnosis. For each group of bearing signals, 100 samples are picked, and the sampling points of each sample are set to 1600. So 1000 samples in total will be selected as experimental verification data. The ratio of the training set to the test set is 3:1.

## 3. Experimental results

### 3.1 Signal decomposition

The VMD parameter optimization is performed according to the description in Section 2.2, and the maximum number of iterations of the DBO algorithm $T_{max}=100$ is set, and the optimization is performed for 10 groups of signal data respectively, and their optimal parameters obtained by the DBO method are shown in Table 3.

Select the F1 data one of the samples as an example, the results of the time-frequency analysis after VMD decomposition are shown in Fig 3 and Fig 4, we can see that the modal aliasing phenomenon and the endpoint effect has a considerable degree of improvement, and decomposition of the modal distinction, indicating that the effect of decomposition is more satisfactory.

In the DBO-VMD decomposition, this research introduces the composite optimization indicator function $L$ instead of the usually used single indicator function as the fitness function. Here use F3 group signals as an example, and the efficacy of the reconstructed signals is compared with that of the single indicator-based fitness function with minimal permutation entropy, kurtosis, and minimum envelope entropy, respectively. The signal-to-noise overlap ratio is used to measure the signal decomposition effect, and the "distorted" component of the decomposition process is treated as noise, therefore the following formula is obtained:

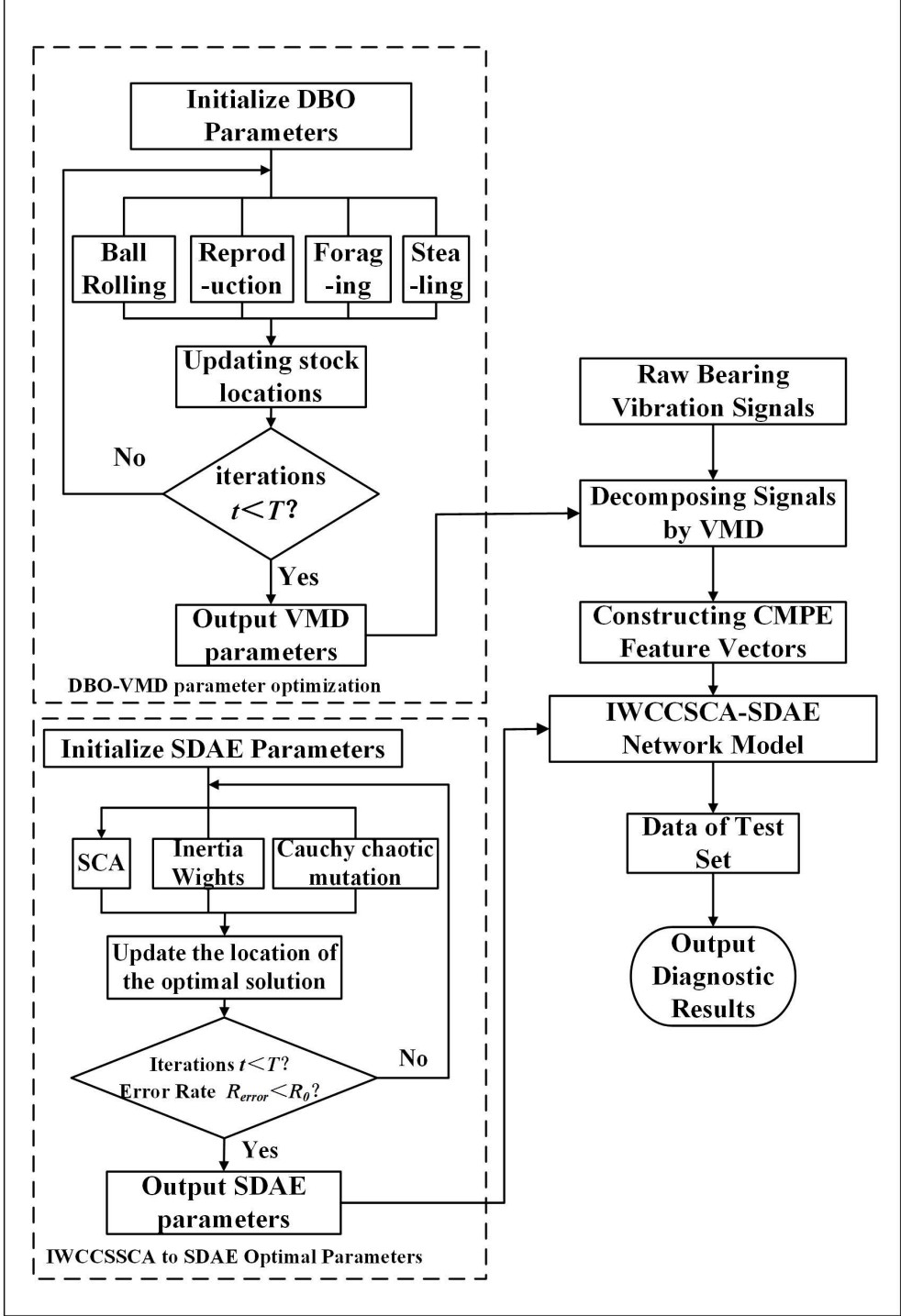

**Fig 2. Diagnosis network model flowchart.**

**Table 2. Bearing fault types and parameters.**

| Tag | Fault type | Size of defect (") | Degree |
|-----|-----------|--------------------|--------|
| F0 | Normal | – | – |
| F1 | Inner race | 0.007 | Slight |
| F2 | Inner race | 0.014 | Medium |
| F3 | Inner race | 0.021 | Severe |
| F4 | Outer race | 0.007 | Slight |
| F5 | Outer race | 0.014 | Medium |
| F6 | Outer race | 0.021 | Severe |
| F7 | Ball | 0.007 | Slight |
| F8 | Ball | 0.014 | Medium |
| F9 | Ball | 0.021 | Severe |

**Table 3. DBO-VMD optimization method parameters.**

| Tag | $K$ – number of IMFs | $\alpha$ – Quadratic penalty Factor |
|-----|---------------------|-------------------------------------|
| F0 | 8 | 2144 |
| F1 | 4 | 2658 |
| F2 | 4 | 2721 |
| F3 | 4 | 1528 |
| F4 | 5 | 2199 |
| F5 | 5 | 1795 |
| F6 | 6 | 1324 |
| F7 | 4 | 1638 |
| F8 | 6 | 1909 |
| F9 | 7 | 2048 |

$$R = 10 * \lg \left[ \left( \frac{\|Y_1\|_2}{\|Y_1 - Y_2\|} \right)^2 \right]$$

(20)

Where $Y_1$ is the reconstructed signal and $Y_2$ is the original signal. Through this equation we may know that the bigger the $R$ value, the less the noise in the reconstructed signal and the more effective information it contains. The SNR metrics obtained by different fitness functions are shown in Table 4, from which we can see that compared with a single metric, the composite optimization metric function as fitness function has a stronger ability to extract effective information, and the SNR recombination degree is improved by 3.39, 6.18, and 8.37 dB compared with the minimum alignment entropy, kurtosis, and minimum envelope entropy, respectively, indicating that the use of composite metrics optimization metric function as the fitness function can effectively improve the feature extraction effectiveness and noise resistance of VMD method.

## 3.2 Fault diagnosis and classification

As can be observed from Step (4) in Section 2.3, for a single set of vibration signal data, the floating threshold approach is employed to complete the normalization of the IMF sample dimensions. From the computational analysis in Table 2, it can be observed that $p = 4$ is adopted as the equivalent number of IMFs. The CMPE feature vector computation is carried out using the approach explained in Section 2.3, appropriate to the experimental circumstance with the aim. In this research,

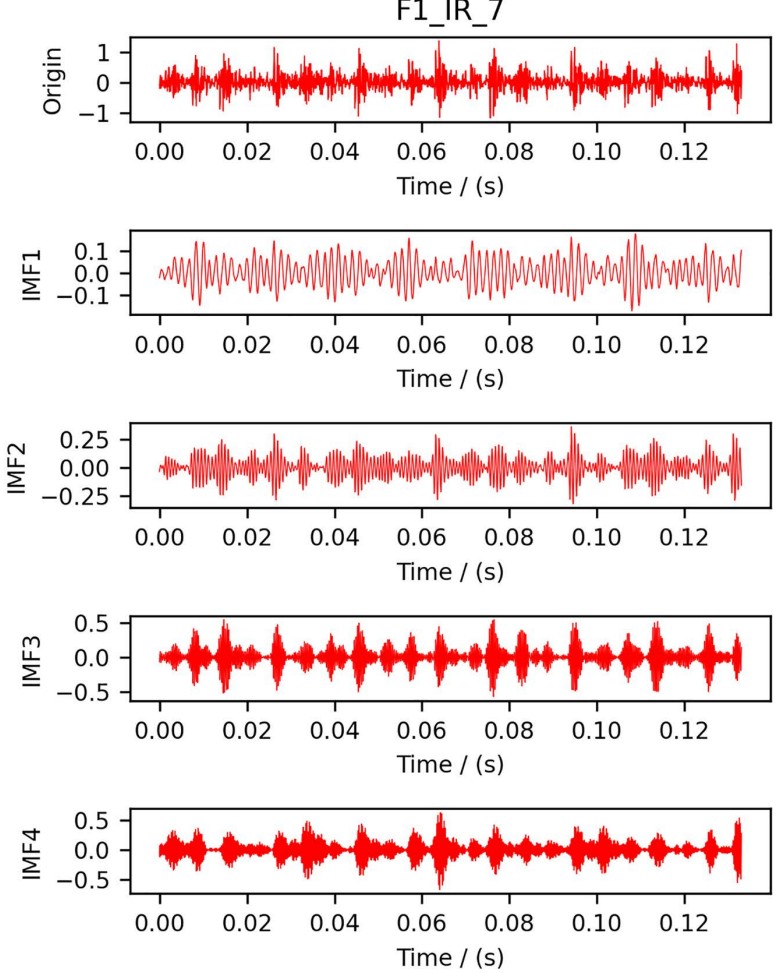

**Fig 3. Time domain diagram after decomposition.**

the embedding dimension size $m = 6$, temporal extension $\lambda = 1$, and scaling factor $\tau = 16$ are chosen. After the calculation, the CMPE feature vector is supplied into the SDAE network model as an input layer.

The parameter settings of IWCCSCA-SDAE correspond to section 2.5, and the SDAE network hyperparameters may be acquired by parameter optimization search, and the particular results are provided in Table 4. For the SDAE network model training parameters, they are set as shown in Table 5. After the setup is completed, the training and testing verification is carried out, the verification process is carried out 50 times, the average of each result is taken for analysis, and the accuracy standard deviation is calculated to analyze the model stability. Table 6 presents the training parameters of the SDAE.

The method used in this paper involves data preprocessing such as signal extraction and feature vector construction, which may increase the computational complexity. Therefore, in order to verify the computational efficiency of our method, we compared it with the following feature extraction-diagnostic classification combination models in terms of model iteration time: VMD-SDAE, EMD-SDAE, and SVM-CNN. The comparative results are presented in the Fig 5.

From the figure, we can observe that owing to the rather extensive data pretreatment method, the first iteration of the technique proposed in this paper is quite sluggish. However, since the data processing results have more obvious

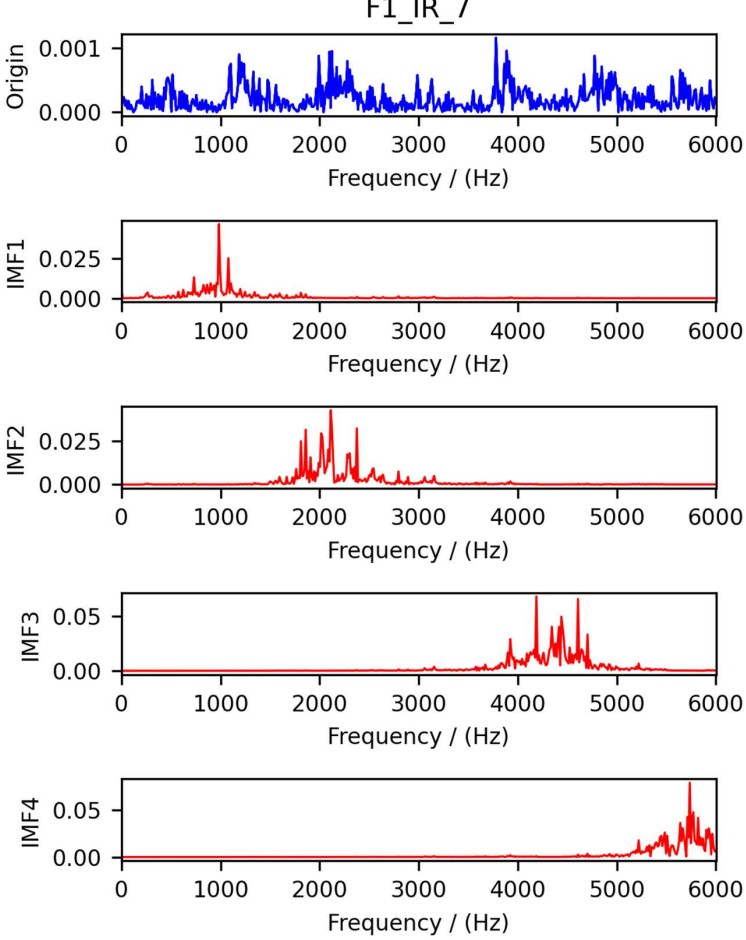

**Fig 4. Frequency domain diagram after decomposition.**

**Table 4. Signal-noise overlap ration comparison.**

| Indicator Type | R/ dB | Relative value | Relative ration |
|---|---|---|---|
| Function *L* | 18.24 | – | – |
| Alignment entropy | 14.85 | −3.39 | −18.59% |
| Envelope entropy | 12.06 | −6.18 | −33.88% |
| Kurtosis | 9.87 | −8.37 | −45.89% |

characteristics, the iteration efficiency can be improved in the subsequent iteration process, thereby alleviating the problem of excessive time spent on data processing in the early stages to a certain extent.

In order to demonstrate the benefit of this paper's technique in terms of convergence speed, diagnostic accuracy, resilience, etc., this paper will include other model as a comparison, and the selection of each algorithm as well as the parameter settings are presented below:

(1) SCA-SDAE: The parameter values are consistent with those in this paper.

**Table 5. SDAE network hyperparameters.**

| Tag | Type of parameters | Value |
|---|---|---|
| $x_{i1}$ | Number of hidden nodes in the first layer | 168 |
| $x_{i2}$ | Number of hidden nodes in the second layer | 254 |
| $x_{i3}$ | Number of hidden nodes in the third layer | 78 |
| $x_{i4}$ | Sparse coefficient of the first layer | 0.1289 |
| $x_{i5}$ | Sparse coefficient of the second layer | 0.2254 |
| $x_{i6}$ | Sparse coefficient of the third layer | 0.1905 |
| $x_{i7}$ | Zero-ratio | 0.05 |

**Table 6. SDAE network training parameters.**

| Parameter type | Value |
|---|---|
| **Number of hidden layers** | 3 |
| **Learning rate** | 0.01 |
| **Batch-size** | 50 |
| **Epoch** | 300 |
| **Activation function** | Sigmoid |

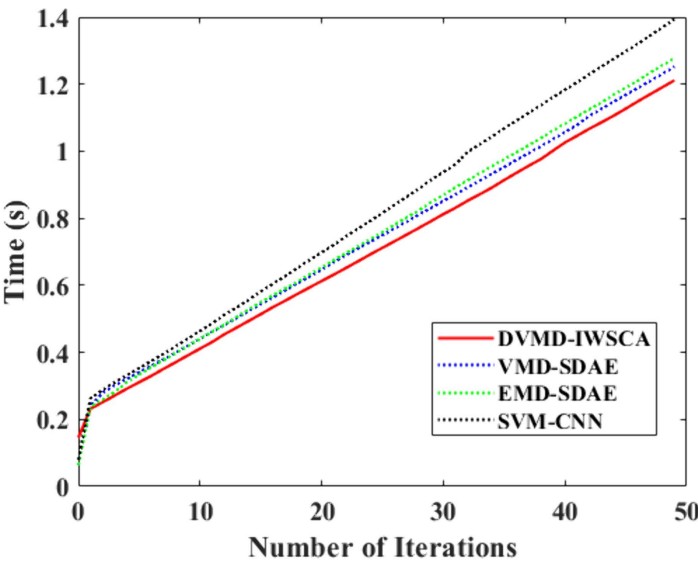

**Fig 5. Time cost comparison.**

(2) PSO-SDAE: In the PSO algorithm, the initialization population size N = 10, maximum number of iterations $T_{max}$ = 100, learning factor c1 = c2 = 2, inertia weights w = 0.4, and parameter settings of the SDAE method are consistent with those of this research.

(3) GA-SDAE: In GA algorithm, the initialized population size N = 50, evolutionary periods T = 200, crossover probability K = 0.8, mutation probability τ = 0.05, and parameter settings of SDAE algorithm are consistent with this research.

(4) SVM: The radial basis kernel function is employed for classification, the penalty factor C = 500, kernel parameters σ = 0.85, and lastly the K-fold cross-validation technique, in the method K = 10.

(5) SVM-CNN: The SVM parameters are configured as stated above. The CNN parameters are as follows: 2 convolution layers, 2 pooling layers, convolution kernel size 5 × 5, pooling layer kernel size 2 × 2, stride 1.

The convergence speed is of significant relevance for the efficiency of the model operation, and the model convergence speed is represented by loss rate curve, and the loss rate convergence graphs of this paper's approach and each comparative method are displayed in Fig 6. From Fig 6, we can see that the IWCCSCA-SDAE method has a faster convergence speed at the beginning, fewer iterations are needed to complete the convergence, and the loss rate is more stable after convergence, which indicates that it has advantages in convergence speed, global optimization search, and local search, and can make the model have a stronger equilibrium search ability and higher running efficiency.

For the fault diagnosis method based on the combination of DBO-VMD and IWCCSCA-SDAE proposed in this paper, the final average diagnostic accuracy reaches 99.51% and the mean square error reaches MSE = $6.6396 \cdot 10^{-6}$, which can be preliminarily seen to have a high diagnostic accuracy after 50 simulation experiments. For the same dataset feature vector input, the diagnosis of 250 samples, a total of 10 types of faults in the test set data, its diagnostic results are shown in Fig 7.

In Fig 7, the red dots show the data with diagnostic mistakes, thus we may derive the following conclusions from Fig 7: the IWCCSCA-SDAE method diagnoses 3 wrong samples with a diagnostic accuracy of 0.988; the SCA-SDAE method diagnoses 14 samples with diagnostic errors with a diagnostic accuracy of 0.944; the PSO-SDAE method diagnoses 20 samples with diagnostic errors with a diagnostic accuracy of 0.920; GA-SDAE method has 17 sample diagnostic errors, with a diagnostic accuracy of 0.932; SVM-CNN method has 22 sample diagnostic errors, with a diagnostic accuracy of 0.912; SVM method has 36 sample diagnostic errors, with a diagnostic accuracy of 0.856. From the above conclusions, we can see that the improved IWCCSCA algorithm for the optimization of hyperparameters of the SDAE network can effectively improve the accuracy of the model fault diagnosis, which has certain advantages compared to the previous methods.

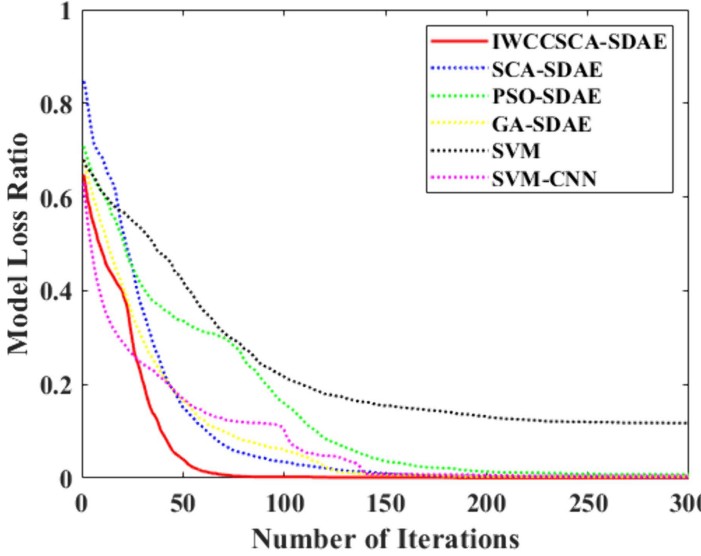

**Fig 6. Loss ratio of models.**

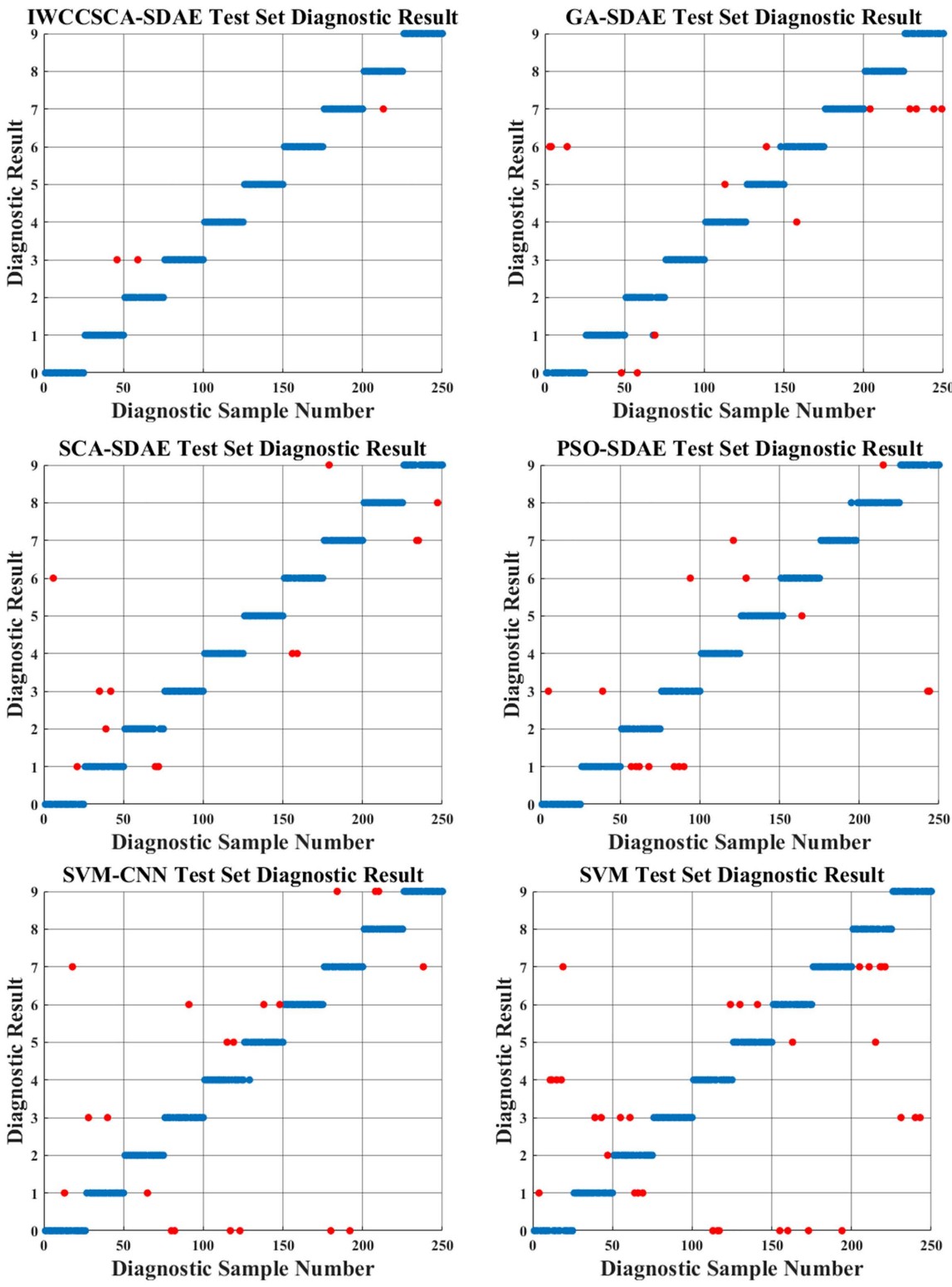

**Fig 7. Model diagnosis results.**

## 3.3 Noise immunity

In order to test the method proposed in this paper in the noise resistance, robustness and other aspects of the enhancement effect, in the original vibration signal to add Gaussian white noise and low-frequency periodic sinusoidal signals, whose frequency is $f=40$ Hz, to simulate the natural environment of the environmental noise and the working environment of the pedestal vibration and other interference signals. A total of five sets of data comprising noise signals are created, and the signal-to-noise ratios are 2dB, 4dB, 6dB, 8dB and 10dB, respectively. The diagnostic accuracy of each model can be obtained by taking the average of diagnostic accuracy of 50 simulation experiments for each method. The comparison graph of each accuracy rate in various noise scenarios is provided in Fig 8.

Through Fig 8, we can see that with the increase of noise intensity, the diagnostic accuracy of each method shows different degrees of diagnostic accuracy decline, but the decline trend of the IWCCSCA-SDAE method is relatively slower, and the average diagnostic accuracy also maintains a higher level compared to other methods, which indicates that its noise resistance and robustness have been improved compared to other methods.

In order to quantify the noise immunity of the method, the noise decrease rate $\alpha_N$ is proposed, which represents the decrease in the diagnostic accuracy of the model compared to the diagnostic accuracy of the un-noised model under specific noise conditions, and the expression is shown in Equation (21).

$$\alpha_N = Accuracy_0 - Accuracy_N \tag{21}$$

Where $Accuracy_0$ represents the diagnostic accuracy of the model for the original signal and $Accuracy_N$ represents the diagnostic accuracy of the model after signal noise addition.

The results of each method and its six groups of anti-noise attenuation rate are shown in Table 7, from which we can easily see that the anti-noise attenuation rate of the IWCCSCA-SDAE method is relatively low, indicating that it is superior in terms of anti-noise, robustness, etc., and also indicating that the addition of the ring noise element in the self-coder network can effectively improve the model anti-noise, robustness, and generalization ability, and improve the diagnostic accuracy of the model in the complex environment. It also indicates that the inclusion of ring noise components to the

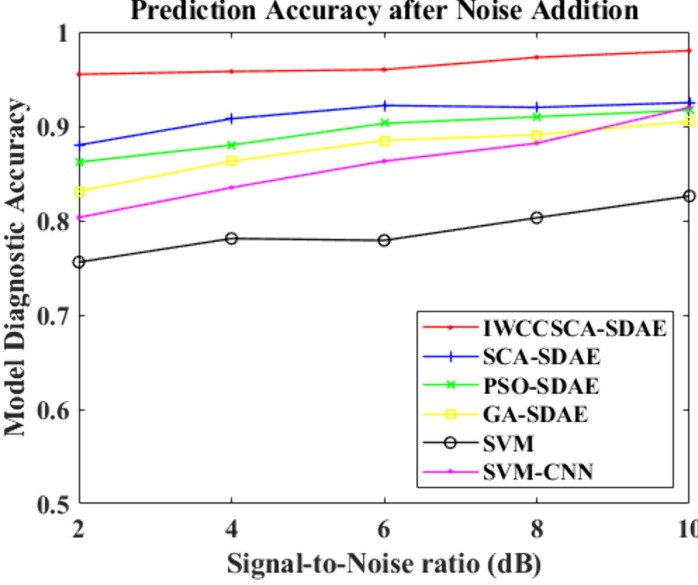

**Fig 8. Diagnosis accuracy of models after adding noise.**

**Table 7. Noise decrease rate comparison.**

| Model type | Noise decrease rate/ % | | | | | Basic rate/ % |
|---|---|---|---|---|---|---|
| | 2dB | 4dB | 6dB | 8dB | 10dB | |
| IWWCCSCA-SDAE | 3.6 | 3 | 2.8 | 1.5 | 0.8 | 98.8 |
| SCA-SDAE | 6.4 | 3.6 | 2.2 | 2.4 | 1.9 | 94.4 |
| PSO-SDAE | 7.4 | 5.6 | 3.3 | 2.6 | 1.9 | 93.6 |
| GA-SDAE | 8.5 | 5.3 | 3.1 | 2.5 | 1.1 | 91.6 |
| SVM-CNN | 10.9 | 7.7 | 4.9 | 3 | 1.0 | 91.2 |
| SVM | 10.0 | 7.5 | 7.7 | 5.3 | 3.0 | 85.6 |

self-encoder network may substantially increase the model's noise resistance, resilience and generalization ability, and improve the model's diagnostic accuracy in complicated situations.

## 4. Discussion and conclusion

Based on the principle of parameter optimization, this research developed a motor bearing fault diagnosis technique based on DBO-VMD with adaptive optimization stacked noise reduction auto encoder.

VMD is a commonly used signal decomposition and feature extraction method, but the effect is affected by the parameter settings, this paper proposed to use the DBO algorithm for parameter optimization of the VMD method under the environment of bearing vibration signals; at the same time, taking into account that the fitness function of a single indicator for the degree of feature extraction needs to be deepened and the generalization ability is insufficient, a composite indicator fitness function $L$ is proposed by combining the Tanimoto coefficients, the permutation entropy values, and the kurtosis. The function $L$ is used as the fitness function of the VMD method to enhance the adaptability and generalizability of the VMD parameters; then the CMPE is used to construct the feature vectors for training the model, and the composite optimization function $L$ is used again as the weighted threshold to control the number of components and the number of feature vectors.

For the SDAE model, a network hyperparameter optimization approach based on IWCCSCA is provided, which offers it with better adaptivity, more balanced global search capacity and local optimization capability, and quicker convergence time. Seven hyperparameters required for the 3-layer SDAE network model are obtained using this approach.

Use the CWRU bearing vibration signal public data set to test the model. Through the tests of signal reduction, diagnostic accuracy, noise immunity and other aspects, we can draw a conclusion that the approach suggested by this research has benefits in the areas of diagnosis accuracy, convergence speed, noise immunity and so on.

However, there still remain certain limits in this strategy. Firstly, the bearing vibration signals used in the experiments are standard data collected in the laboratory. Extracting a vast quantity of clear vibration signals is challenging in real work settings of bearing. Therefore, in the case of interference in the signal extracting process, how to complete the feature extraction, data enhancement and classification based on the acquired data is the future research direction.

Besides, the function $L$ used in the DBO-VMD method is verified to be validated under the conditions of experiment of this paper, but the validity of the function under other types of conditions remains to be verified, and the parameters and structure of the function can be further refined through further research and experiment, to accommodate different types of fault diagnosis.

## Author contributions

**Conceptualization:** Haowen Li.

**Data curation:** Laixian Chen.

**Funding acquisition:** Xianlin Ren.

**Methodology:** Siyao Xiong.

**Software:** Zhengwen Li.

**Writing – original draft:** Haowen Li.

**Writing – review & editing:** Xianlin Ren.

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
