## [Decision Letter · Decision Letter 0]

19 May 2025

Dear Dr. Li,

Thank you for submitting your manuscript to PLOS ONE. After careful consideration, we feel that it has merit but does not fully meet PLOS ONE’s publication criteria as it currently stands. Therefore, we invite you to submit a revised version of the manuscript that addresses the points raised during the review process.

ACADEMIC EDITOR: Please address all the comments through a revised manuscript and a separate response letter.  

We look forward to receiving your revised manuscript.

Kind regards,

Qichun Zhang, PhD

Academic Editor

PLOS ONE

Journal Requirements:

3. Thank you for stating the following financial disclosure: [The research is supported by the National Natural Science Foundation of China�51875090]. 

4. Thank you for uploading your study's underlying data set. Unfortunately, the repository you have noted in your Data Availability statement does not qualify as an acceptable data repository according to PLOS's standards.

Additional Editor Comments:

Basically, this paper seems like a converted version of one MSc dissertation. VMD, DBO, iterative optimisation etic. have been combined together to improve the diagnosis performance. Technical speaking, the approach is sound while it also lacks of novelty. As a few writing problems are noticed, I recommend a major revision before the re-evaluation. The following comments may be helpful to enhance the quality of the paper.

1) Please highlight the innovations and novelties of the paper.

2) Please indicate the motivation of the research and the methodologies' combination. Why these methods can be combined and what is the benefit for this type of combination.

3) The main challenge of bearing fault diagnosis has not been given clearly.

4) Some standard method such as VMD is well-established, the explanation can be simplified to shorten the paper.

5) Nearly all the formula needs to be updated including all the symbols in the equations should be pre-defined clearly before usage.

6) A pseudocode needs to be given to show the implementation flow of the proposed method.

7) Please explain the source of the experimental data, is the dataset a benchmark?

8) A proof reading is highly recommended as the current typos and gramma problem affect the readability of the paper.

Reviewers' comments:

Reviewer's Responses to Questions

**Comments to the Author**

1. Is the manuscript technically sound, and do the data support the conclusions?

Reviewer #1: Yes

2. Has the statistical analysis been performed appropriately and rigorously?

Reviewer #1: Yes

3. Have the authors made all data underlying the findings in their manuscript fully available?

Reviewer #1: Yes

4. Is the manuscript presented in an intelligible fashion and written in standard English?

Reviewer #1: Yes

Reviewer #1: 1.Innovative expressions should be clear to avoid merely piling up methods. The paper combines various algorithms, but lacks sufficient justification for why these methods were chosen. The need for each method and how they work together to address the core challenge of rolling bearing fault diagnosis needs to be clearly stated.

2.Tanimoto coefficients are commonly used for set similarity comparison, whereas permutation entropy and kurtosis reflect signal complexity and shock characteristics. However, the physical meaning and mathematical compatibility of combining these three as fitness functions have not been fully explained. It is recommended to verify the contribution of each index within the fitness function through an ablation experiment or to provide a theoretical basis.

3.Hyperparameter sensitivity experiments need to be supplemented to evaluate the effectiveness of the improvement.

4.There is insufficient experimental comparison. The current mainstream methods, such as deep convolutional networks, are not clearly compared.

5.Computational costs are not discussed. The method involves multiple optimization and feature extraction, which may have high computational complexity. The time cost needs to be analyzed compared to a lightweight model.

**Do you want your identity to be public for this peer review?** For information about this choice, including consent withdrawal, please see our Privacy Policy

Reviewer #1: No

---

## [Author Response · Author response to Decision Letter 1]

18 Sep 2025

Dear Editor and Reviewers:

Thank you for your letter and for the reviewers’ comments concerning our manuscript entitled “Bearing fault diagnosis method based on enhanced VMD and adaptive-optimized SDAE”. Those comments are valuable and very helpful for revising and improving our paper, as well as the important guiding significance to our researches. We have studied comments carefully and have made corrections which we hope meet with your approval.

In the file named 'Response to Reviewers' are our detailed responses to each comment and the detail revisions are in the revised manuscript. We hope that the revisions in the manuscript and our accompanying responses will be sufficient to make our manuscript suitable for publication in PlOS ONE.

We shall look forward to hearing from you at your earliest convenience.

Sincerely yours!

Haowen Li

University of Electronic Science and Technology of China

202322040545@std.uestc.edu.cn

---

## [Editor Report · Decision Letter 1]

14 Nov 2025

Bearing fault diagnosis method based on enhanced VMD and adaptive-optimized SDAE

PONE-D-25-14415R1

Dear Dr. Li,

We’re pleased to inform you that your manuscript has been judged scientifically suitable for publication and will be formally accepted for publication once it meets all outstanding technical requirements.

Kind regards,

Qichun Zhang, PhD

Academic Editor

PLOS ONE

Additional Editor Comments:

Since the reviewer is not available for re-reviewing this submission. The academic editor has to be an additional reviewer for evaluating this manuscript and the revision. After going through the revised version and the responses to the comments, I believe that all the concerns have been addressed well in the current revised version while the contribution has been highlighted clearly. Based on the quality and improved readability, I recommend accepting this paper as it is which is ready for publication.

---

## [Editor Report · Acceptance letter]

PONE-D-25-14415R1

PLOS ONE

Dear Dr. Li,

I'm pleased to inform you that your manuscript has been deemed suitable for publication in PLOS ONE. Congratulations! Your manuscript is now being handed over to our production team.

Kind regards,

on behalf of

Prof. Qichun Zhang

Academic Editor

PLOS ONE